# Prevalence of Mental Health Disorders among Elderly Diabetics and Associated Risk Factors in Indonesia

**DOI:** 10.3390/ijerph181910301

**Published:** 2021-09-30

**Authors:** Mahalul Azam, Rina Sulistiana, Arulita Ika Fibriana, Soesmeyka Savitri, Syed Mohamed Aljunid

**Affiliations:** 1Department of Public Health, Faculty of Sports Science, Universitas Negeri Semarang, Semarang 50229, Indonesia; rinasulistiana25@gmail.com (R.S.); arulita.ika.f@mail.unnes.ac.id (A.I.F.); 2Department of Psychiatry, Dr. Kariadi General Hospital, Semarang 50244, Indonesia; meykadidit@gmail.com; 3Department of Health Policy and Management, Faculty of Public Health, Kuwait University, Kuwait City 11311, Kuwait; saljunid@gmail.com

**Keywords:** mental health, diabetes, Indonesia, risk factors, elderly

## Abstract

This cross-sectional study aimed to explore mental health disorders (MHD) prevalence among elderly diabetics in Indonesia. Data were extracted from the 2018 national basic health survey in Indonesia (abbreviated as RISKESDAS). The survey involved households randomly selected from 34 provinces, 416 districts, and 98 cities in Indonesia, with 1,017,290 respondents. The number of subjects selected in this study was 2818 elderly diabetic subjects. MHD was determined by self-reporting assessment. Secondary data acquired from RISKESDAS 2018 data involved age, sex, urban–rural residence status, marital status, educational level, employment status, obesity, hypertension, heart disease, stroke, family history of MHD, and DM duration. Binary logistic regression with a backward stepwise method was used to analyze the risk factors related to MHD. MHD prevalence among elderly diabetics in Indonesia was 19.3%. Factors associated with MHD among elderly diabetics were being female (prevalence odds ratio (POR) = 1.64; 95% CI: 1.126–2.394), married (POR = 0.05; 95% CI: 0.031–0.084), less education (POR = 3.37; 95% CI: 1.598–10.355), and stroke (POR = 1.61; 95% CI: 1.183–2.269). MHD prevalence among elderly diabetics in Indonesia was 19.3%, suggesting that screening for psychological problems and educating elderly diabetic patients is essential. Unmarried female elderly diabetics with less education and stroke were altogether more likely to experience MHD.

## 1. Introduction

In 2019, it was reported that 463 million individuals globally suffered from diabetes mellitus (DM). This number has increased from 382 million in 2013 [1]. The United States, China, India, and Indonesia are countries with a high prevalence of DM [2]. The prevalence of DM in Indonesia was 5.7% in 2007, which increased by 10.9% in 2018 [3,4], representing 157,500, or 6%, of total deaths [5]. In 2019, DM represented a catastrophic disease and a financial burden that cost USD 381.25 million in hospital treatment, based on national health insurance (Jaminan Kesehatan Nasional = JKN) [6].

Mental health disorder (MHD) is a common comorbidity in DM, with a prevalence of 28% globally; females tend to suffer more than males, i.e., 34% and 23%, respectively [7,8,9,10]. Mental disorders such as generalized anxiety disorder (GAD), major depressive disorder (MDD), bipolar disorder, and eating disorders are common in DM patients [11,12,13,14]. MHD in diabetics may decrease quality of life [15] and poor self-care management [16], as well as increase disability [17], cardiovascular mortality risk [18] and the risk of all-cause mortality [19]. On the other hand, diabetes is a risk factor for MHD [20]. In general population studies, younger diabetics are more likely to develop MHD [8]. Another study reported that elderly diabetics are more likely to suffer from MHD, with an increased risk for other factors [11]. A previous study also concluded that MHD is more likely to occur in females with no formal education, current alcohol abusers, those with type 1 DM, a longer duration of DM, a chronic complication of DM, and other comorbidities common among elderly diabetic patients [21]. Previous studies concern the association of MHD diabetic comorbidity with genetic and family history [22,23,24,25] as well as obesity [26,27,28].

The frequent coexistence of mental health conditions in elderly diabetics should be of concern [29]. The mechanisms of psychiatric illness involving brain-derived neurotrophic factors, insulin resistance, and inflammatory cytokines could be due to the pathogenesis of DM and several psychiatric illnesses in the elderly [29]. Physical and psychosocial changes affect both mental health and diabetes in the elderly [30]. Diabetic complications such as retinopathy, nephropathy, neuropathy, coronary artery disease, and cerebrovascular disease were also associated with poor mental health status in elderly diabetics [31]. Another study also concluded that overweight status, poor physical capabilities, low activity level, and diabetic complications were risk factors for depression in elderly diabetic patients [32]. However, there is a lack of information regarding MHD risk factors among elderly diabetics in Indonesia. The five-annual national basic health survey (abbreviated as an acronym of RISKESDAS: riset kesehatan dasar) 2018 [3] was the latest national survey conducted by the Ministry of Health, Republic of Indonesia. The present study aims to determine the prevalence and risk factors of MHD among elderly diabetics in Indonesia.

## 2. Materials and Methods

### 2.1. Design and Study Population

This cross-sectional study employed secondary data acquired from RISKESDAS 2018, which is the latest round of the study. The survey involved households randomly selected from 34 provinces, 416 districts, and 98 cities in Indonesia, with 1,017,290 respondents [3]. The study population involved elderly diabetics older than 60. Diabetic status was determined by fasting blood glucose level ≥ 126 mg/dL, or 2 h postprandial and random blood glucose level ≥ 200 mg/dL, or that which had previously been diagnosed by a doctor. Blood glucose levels were measured using Accu-Check Performa (Roche, Basel, Switzerland). Subjects with incomplete data were excluded from the study. Details of data collection, ethical issues, and other related steps were published in the RISKESDAS 2018 report [3].

### 2.2. Data Collection

This study was approved by the Ethics Committee, the National Institute of Health Research and Development (NIHRD), the Ministry of Health, Republic, Indonesia. MHD status was determined by a WHO self-reporting questionnaire-20 (SRQ-20), [33,34,35] as acquired from RISKESDAS 2018 data (Appendix A). SRQ-20 is a tool used to measure common mental disorder symptoms [33,34,35]. The SRQ-20 consists of 20 questions regarding the prevalence of somatic, cognitive, and emotional symptoms over the past 30 days: 0 = No and 1 = Yes [33,34,35]. RISKESDAS 2018 refers to a previous study that validated SRQ-20 in the Indonesian population [35]. The study determined MHD with the cut-off point ≥ 6, positive predictive value = 70%, and negative predictive value = 92% [35]. Secondary data were also acquired from RISKESDAS 2018 that involved age, sex, urban rural residence status, marital status, educational level, employment status, obesity, hypertension, heart disease, stroke, family history of MHD, and duration of DM.

### 2.3. Statistical Analysis

Subjects’ characteristics were presented as frequency and proportions. The relationships of the determinants and MHD status were analyzed by a chi-square test. The *p*-values < 0.05 were considered statistically significant. Binary logistic regression with a backward elimination (conditional) method was conducted to acquire the regression model since the dependent variable scale was nominal. The dependent variable was MHD status that was categorized as “Yes = 1” if it meets the criterion and “No = 1” if not. We presented a prevalence odds ratio (POR) for cross-sectional study as formulated in the previous study [36]. All statistical analyses were performed using Statistical Package for the Social Sciences (SPSS) software (version 23.0 for Windows, IBM SPSS Inc., Chicago, IL, USA).

## 3. Results

Data extracted from RISKESDAS 2018 contained 2818 elderly diabetic subjects. Table 1 shows that the proportion of female elderly diabetics in the study population was higher, while the age category is almost comparable. Most elderly diabetics had less education, lived in an urban area, were unemployed, and were married. A small number of the study population, i.e., around 6%, was obese and had a family history of MHD. Hypertension and duration of DM were almost comparable, while heart disease and stroke had a lower proportion in the total population. The overall prevalence of MHD among elderly diabetics was 19.3% in the study population.

Table 2 identifies variables related to MHD. Sex, residence type, educational level, employment status, obesity, hypertension, heart disease, stroke, and family history of MHD were significantly different between the MHD groups based on the chi-square test. However, age, marital status, and duration of DM were comparable between the groups. The proportions of several parameters were significantly higher in the MHD group, i.e., female, rural residence, lower educational level, unemployed, obesity, hypertension, heart disease, family history of MHD, and stroke.

Backward (conditional) binary logistic regression results concluded the final model of regression shown in Table 3. The final model concluded that being female (POR = 1.64; 95% CI: 1.126–2.394), having less education (POR = 3.73; 95% CI: 1.598–10.355), and stroke (POR = 1.61; 95% CI: 1.61) were altogether associated with MHD among elderly diabetics.

## 4. Discussion

This cross-sectional study involved 2818 elderly diabetics in Indonesia. Of them, 545 experienced MHD, indicating that the prevalence of MHD among elderly diabetics in this study population was 19.3%. The current study updated the prevalence of MHD among elderly diabetics aged older than 60 years old, especially in Indonesia. A systematic review involving 248 studies estimated a prevalence of 28% of people with type 2 diabetes who experienced depression globally, with 32% being in Asia [8]. People with diabetes aged older than 65 years old had a prevalence ratio of 21% [8], which is a similar value to that indicated in the current study, while those of a younger age (<65 years old) had a greater prevalence ratio, i.e., 31% [8]. The female group had more prevalence than the male group, i.e., 34% and 24%, respectively [8]. Depression determination methods also influence the prevalence ratio; self-reported methods tend to have a higher prevalence (30%) than clinical diagnosis assessment (22%) [8]. The current study utilized self-reported methods using WHO-SRQ-20 [33,34,35]; however, it found a lower prevalence than the previous review [8]. A previous systematic review of 26 studies involving all measurement assessments conducted in 2011 concluded that the prevalence of major depressive disorder in type 2 DM was 14.5%, indicating a lower prevalence ratio [37]. Another study observed diabetic patients aged over 55 years in primary care and found an MHD prevalence of 19.1% [38].

Evidence shows that diabetes mellitus is reciprocally associated with MHD and coincides as a comorbidity [39]. Depression is a common MHD that is discussed as a risk factor of DM [20]; however, the underlying mechanism is still unclear. Chronic stress induces immune dysfunction through the hypothalamus-pituitary-adrenal axis and the sympathetic nervous system, causing hypercortisolemia and promoting insulin resistance and visceral obesity, and leading to metabolic syndrome and DM [39]. Furthermore, chronic stress increases the production of inflammatory cytokines. High inflammatory cytokines interact with the function of pancreatic β-cells, induce insulin resistance, and promote the appearance of type 2 diabetes mellitus [39]. On the other hand, pro-inflammatory cytokines have been reported to influence pathophysiological domains that characterize depression, including neurotransmitter metabolism, neuroendocrine function, synaptic plasticity, and behavior [40]. This association suggests that both stress and inflammation stimulate depression and diabetes mellitus [39]. Chronic stress and inflammation processes, as well as the physical and psychosocial changes that are common in the elderly population, affect both mental health and diabetes in the elderly [30].

The present study found that having a lower educational level, being female, being unmarried, and stroke were all associated with MHD among elderly diabetics with the pseudo-R-squared (Nagelkerke) value being 0.790. This finding explained that 79.0% of MHD determinants in this population study of elderly diabetics were influenced by the mentioned factors. The rest, 21.0%, can be explained by other factors that are not observed in the study. The present study involved many determinants, but only those provided in the RISKESDAS 2018 data. This study did not observe other pivotal determinants for MHD in people with diabetes. The determinants involved physical capability, insulin and drug usage, ethnicity, detailed civil status (married, single, divorced, widowed), residence status (living alone, nuclear family, joint/extended living), family size, family income, pensioner status, smoking, alcohol, religion, glycemic control, and other sociodemographic and clinical health factors [8,9,11,32,41,42]. The involved determinants in the study can potentially explain other parameters that influence MHD among elderly diabetics.

Lower education level was a factor that was significantly associated with MHD in this study. Many other studies have reported a similar association of lower educational status with common mental disorders in the diabetic population as well as in the general population [43,44,45,46]. Lower education level diabetics have limitations in coping [47] with diabetes complications and other comorbidities, as well as general psychosocial problems. This study showed that elderly diabetics with lower education had more than three times the risk of experiencing MHD compared to those with more education. The present study determined lower education levels for elderly diabetics as being those who had passed junior high school (secondary education) or a lower form education. The other study categorized education level in greater detail, i.e., no schooling, primary education, secondary education, and tertiary education [41].

This study also concluded that female elderly diabetics had a 64% higher risk of acquiring MHD than males. Females are more likely to acquire MHD in the general population, as well as among diabetics and chronic disease patients [8,48]. The current study also demonstrated a significant relationship between stroke and MHD. The risk of stroke is 61% higher compared to the absence of stroke. The presence of stroke and diabetes is often followed by other multimorbidities, including MHD [49]. Some studies have also concluded that the presence of a comorbidity was more likely to initiate MHD [11]. The more comorbidities and additional illnesses, the higher the risk of acquiring MHD [11]. This condition is also related to physical limitations and capabilities, as well as the complicated clinical and health conditions involved in drug use [32].

A limitation of our study is the absence of diabetic medication status and glycemic control [29,32]. Glycemic control and the use of certain oral medication are related to mental health conditions in elderly diabetics [29,32]. Previous studies involved stress and epigenetics as predictors of MHD in the general population [23]. Candidate genes were also studied and revealed that APOE, BDNF, and SLC6A4 polymorphisms were related to MHD in the general population [24]. Other studies revealed the contribution of inflammatory markers to mental health. Interleukin (IL)-1β, IL-6, IL-10, monocyte chemoattractant protein-1, tumor necrosis factor-alpha, C-reactive protein, and phospholipase A2 contribute to depression [25], which is also associated with type 2 diabetes [50]. However, our study did not involve biological and genetic markers to elucidate an understanding of these mechanisms. Furthermore, regarding the statistical analysis, a backward stepwise binary logistic regression was chosen as an efficient method for the extensive data; however, there are some restrictions regarding this method [51].

## 5. Conclusions

The prevalence of MHD among elderly diabetics in Indonesia was 19.3%. The risk factors for MHD among elderly diabetic subjects were female, no marriage, low education, and stroke. The high prevalence of MHD among elderly diabetics suggests that screening for psychological problems and educating elderly diabetic patients should be considered as routine components for diabetes care. Further studies should be conducted using clinical diagnostic assessments in a large population study involving genetic factors, inflammatory markers, cardiometabolic traits, and other potential factors in order to elucidate the relationship between risk factors and the occurrence of MHD among elderly diabetic subjects, as well as their mechanisms.

## Figures and Tables

**Table 1 ijerph-18-10301-t001:** Subjects’ characteristics.

Characteristics	Frequency (n)	Percentage (%)
Sex		
Male	1163	41.3%
Female	1655	58.7%
Age		
60–69 years	1489	52.8%
>69 years	1329	47.2%
Education level		
JHS or lower	2049	72.7%
SHS or higher	769	27.3%
Residence		
Urban	1799	63.8%
Rural	1099	36.2%
Employment status		
Unemployed	1723	61.1%
Employed	1095	38.9%
Marital status		
Not married	1182	41.9%
Married	1636	48.1%
Obesity		
Yes	189	6.7%
No	2629	93.3%
Family history of MHD		
Yes	171	6.1%
No	2647	93.9%
Duration of DM		
≤5 years	1555	55.2%
≥5 years	1263	44.8%
Hypertension		
Yes	1441	51.1%
No	1377	48.9%
Heart diseases		
Yes	347	12.3%
No	2471	87.7%
Stroke		
Yes	258	9.2%
No	2560	91.8%
MHD		
Yes	545	19.3%
No	2.273	80.7%

JHS: junior high school, SHS: senior high school.

**Table 2 ijerph-18-10301-t002:** Subjects’ characteristics based on MHD status.

Variables	MHD				*p*-Value *	POR	% CI
	Yes (n = 545)		No (n = 2273)				
	n	%	n	%			
Age Category					0.11	1.17	0.968–1.407
60–69 years	271	18.2	1218	81.8			
>69 years	274	20.6	1055	79.4			
Sex					0.001	1.56	1.282–1.901
Female	366	22.1	1289	77.9			
Male	179	15.4	984	84.6			
Residence					0.001	0.61	0.504–0.736
Urban	296	16.5	1503	83.5			
Rural	249	24.4	770	75.6			
Marital Status					0.08	1.19	0.983–1.432
Not married	247	20.9	935	79.1			
Married	298	18.2	1338	81.8			
Educational Level					0.001	2.49	1.932–3.200
JHS or lower	464	22.6	1585	77.4			
SHS or higher	81	10.5	688	89.5			
Employment Status					0.003	1.35	1.111–1.647
Unemployed	364	21.1	1359	78.9			
Employed	181	16.5	914	83.5			
Obesity					0.001	5.51	4.072–7.468
Yes	100	52.9	89	47.1			
No	445	16.9	2184	83.1			
Hypertension					0.001	1.96	1.618–2.383
Yes	351	24.3	1090	75.6			
No	194	14.0	1183	85			
Heart Diseases					0.02	1.37	1.047–1.785
Yes	83	23.9	264	76.1			
No	462	18.7	2009	81.3			
Stroke					0.001	2.29	1.733–3.022
Yes	86	33.3	172	66.4			
No	495	19	2101	81.0			
Family History of MHD					0.001	2.62	1.891–3.630
Yes	63	36.8	108	63.2			
No	482	18.2	2165	81.8			
Duration of DM					0.94	0.99	0.823–1.199
≤5 years	245	19.4	1018	80.6			
≥5 years	300	19.3	1255	80.7			

JHS: junior high school, SHS: senior high school, POR: prevalence odds ratio. * Chi-square test.

**Table 3 ijerph-18-10301-t003:** Binary logistics regression of risk factors for MHD.

Risk Factors	B	*p*-Value	POR	95% CI
Female	0.496	0.01	1.64	1.126–2.394
Married	−2.979	0.001	0.05	0.031–0.084
Lower education	3.518	0.001	3.73	1.598–10.355
Stroke	0.480	0.049	1.61	1.183–2.659
Constant	−3.847			

Nagelkerke pseudo-R-squared: 0.790; POR: prevalence odds ratio.

## Data Availability

The data used to support the findings of this study are available from the corresponding author Mahalul Azam upon request through the email address mahalul.azam@mail.unnes.ac.id.

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
