# Peer review of "Prevalence of Mental Health Disorders among Elderly Diabetics and Associated Risk Factors in Indonesia"

_ijerph, 2021, doi:10.3390/ijerph181910301_

Round 1

Reviewer 1 Report

Summary:

The paper presents an application on data collected from a cross-sectional national basic health survey (RISKESDAS 2018), involving a subset of 2,818 Indonesian elderly diabetic subjects, to determine the prevalence and risk factors of mental health disorders (MHD) among elderly diabetics in Indonesia.

In order to pursue their goals, the authors performed a binary logistic regression, after performing the Chi2 test to select the predictors of the regression model.

Some Remarks:

  1. The procedure used for the selection of predictors is not adequate. Specifically, the authors select those predictors that in the Chi2 test with the MHD variable, have a p-value <0.025.

Those p-values are not a measure of the importance or significance of variables.

However, since a p-value less than 0.025 indicates strong evidence against the null hypothesis, the criterion could be applied for screening in first place predictors. However, it remains opportune to test the significance of the predictors in explaining the response through the model.

My suggestion is to perform stepwise model selection in order to define the predictors of the logistic regression.

  1. On page 5 (line 135) authors present the overall model fit index, Nagelkerke's pseudo R-squared (not pseudo R). The pseudo R-squared value leaves something to be desired. It will probably be worth the effort to revise the model to try to make better predictions.

  1. The literature review is poor and not entirely adequate to support the relationships among the collected data.

My suggestion is to revise the theoretical framework by carefully considering authors’ research question.

  1. It is not clear to me why the authors support their use of the SRQ-20 scale by resorting on the fact that it has been validated in Eritrea (Reference n. 22 cited in Materials and Methods in the page 2, row 70).

  1. The following key aspects of the SRQ-20 scale are not sufficiently clear in the manuscript:

- what types of mental disorders the SRQ-20 scale is able to screen;

- what is the cut-off value for which the subjects were classified as suffering from MHD;

- if the SRQ-20 scale has been validated in Indonesia;

- what is the positive predictive value or the negative predictive value.

  1. Finally, in the "Discussion" section, I suggest that the authors should also report the results obtained from the estimated logistic regression model (and not just refer the reader to the odds ratios shown in Table 3), in order to offer to IJERPH’s readers a comprehensive understanding of the analysed data.

To be clear and concise: I recommend adopting a clear analysis strategy and being rigorous in its application and comprehensible in its description.

Author Response

Response to Reviewer 1 Comments

General Comment

The paper presents an application on data collected from a cross-sectional national basic health survey (RISKESDAS 2018), involving a subset of 2,818 Indonesian elderly diabetic subjects, to determine the prevalence and risk factors of mental health disorders (MHD) among elderly diabetics in Indonesia.

In order to pursue their goals, the authors performed a binary logistic regression, after performing the Chi2 test to select the predictors of the regression model.

Response:

We thank the editor and reviewers that provide us the importantly substantial review and inputs. We believe these reviews will improve our manuscript; therefore, we are trying the best to revise.

Point 1

The procedure used for the selection of predictors is not adequate. Specifically, the authors select those predictors that in the Chi2 test with the MHD variable, have a p-value <0.025.

Those p-values are not a measure of the importance or significance of variables.

However, since a p-value less than 0.025 indicates strong evidence against the null hypothesis, the criterion could be applied for screening in first place predictors. However, it remains opportune to test the significance of the predictors in explaining the response through the model.

My suggestion is to perform stepwise model selection in order to define the predictors of the logistic regression.

Response 1

We decided to conduct re-analysis using backward (conditional) stepwise Binary logistic since the mis-determination of candidate variable that have p value of <0.25.

The statements provide in page 3 line 106-107

Binary logistic regression with the Backward Elimination (Conditional) method was conducted to acquire the regression model.

Point 2

On page 5 (line 135) authors present the overall model fit index, Nagelkerke's pseudo R-squared (not pseudo R). The pseudo R-squared value leaves something to be desired. It will probably be worth the effort to revise the model to try to make better predictions.

Response 2

We preform repeat analysis using backward (conditional) stepwise Binary logistic and found changes in pseudo-R-squared = 0.790

Page 5, line 138 Table 3. Binary logistics regression of risk factors for MHD and paragraph in line 139-147

Backward (Conditional) Binary logistic regression results concluded the final model of regression showed in Table 3. The final model concluded that female (prevalence odds ratio [POR]=1.64; 95% CI: 1.126-2.394), lower education (POR=33.73; 95% CI: 10.598-107.355), stroke (POR=1.61; 95% CI: 1.61), altogether were associated with MHD among elderly diabetics.

.

Point 3

The literature review is poor and not entirely adequate to support the relationships among the collected data.

My suggestion is to revise the theoretical framework by carefully considering authors’ research question.

Response 3

We add some information regarding MHD as a common mental health that screened by SRQ-20. (Page 2 line 94-95)

SRQ-20 is the tool that was used to measure symptoms of common mental disorder[22, 23].

Some not adequate references that did not support the relationship among collected data, especially in the expnataion of relationship of determinants: obesity and family history of MHD which are low evidence of relaionship in the previous studies, were excluded from the discussion.

In the background section we add some information to streghten the theoretichal framework (Page 2 lin 55-68)

The frequent coexistence of mental health conditions in elderly diabetics should be of concern[29]. The mechanisms of psychiatric illness involving brain-derived neurotrophic factors, insulin resistance, and inflammatory cytokines could be due to the pathogenesis of DM and several psychiatric illnesses in the elderly[29]. Physical and psychosocial changes affect both mental health and diabetes in the elderly[30]. Diabetic complications such as retinopathy, nephropathy, neuropathy, coronary artery disease, and cerebrovascular disease were also associated with the poor mental health status in elderly diabetics [31]. The other study also concluded that overweight status, poor physical capabilities, low activity level, and diabetic complications were risk factors for depression in elderly diabetic patients[32].

Point 4

It is not clear to me why the authors support their use of the SRQ-20 scale by resorting on the fact that it has been validated in Eritrea (Reference n. 22 cited in Materials and Methods in the page 2, row 70).

Response 4

We changed this reference to previous related studies in Indonesia (reference no 33-35)

Reuter A, Vollmer S, Aiyub A, Susanti SS, Marthoenis M (2020) Mental distress and its association with sociodemographic and economic  characteristics: community-based household survey in Aceh, Indonesia. BJPsych open 6:e134

Irmansyah I, Dharmono S, Maramis A, Minas H. Determinants of psychological morbidity in survivors of the earthquake and tsunami in Aceh and Nias. Int J Ment Health Syst. 2010 Apr 27;4(1):8. doi: 10.1186/1752-4458-4-8. PMID: 20423505; PMCID: PMC2873571.

Ganihartono I (1996) PSYCHIATRIC MORBIDITY AMONG PATIENTS ATTENDING THE BANGETAYU COMMUNITY HEALTH CENTRE IN INDONESIA. Bul. Penelit. Kesehat. 24:

Point 5

The following key aspects of the SRQ-20 scale are not sufficiently clear in the manuscript:

- what types of mental disorders the SRQ-20 scale is able to screen;

- what is the cut-off value for which the subjects were classified as suffering from MHD;

- if the SRQ-20 scale has been validated in Indonesia;

- what is the positive predictive value or the negative predictive value.

 Response 5

- what types of mental disorders the SRQ-20 scale is able to screen;

SRQ-20 screened the prevalence of somatic, cognitive, and emotional symptoms over the past 30 days

- what is the cut-off value for which the subjects were classified as suffering from MHD;

This study determined MHD with the cut-off point ≥6; positive predictive value=70% and negative predictive value=92%[24]

- if the SRQ-20 scale has been validated in Indonesia;

RISKESDAS 2018 refers to the previous study that validated SRQ-20 in the Indonesian population[24]

- what is the positive predictive value or the negative predictive value.

This study determined MHD with the cut-off point ≥6; positive predictive value=70% and negative predictive value=92%[24]

We revised it in page 2-3 line 94-99

SRQ-20 is the tool that was used to measure symptoms of common mental disorder[22–24]. SRQ-20 consists of 20 questions regarding the prevalence of somatic, cognitive, and emotional symptoms over the past 30 days, measure 0=No and 1=Yes[22–24]. RISKESDAS 2018 refers to the previous study that validated SRQ-20 in the Indonesian population[24]. This study determined MHD with the cut-off point ≥6; positive predictive value=70% and negative predictive value=92%[24]

Point 6

Finally, in the "Discussion" section, I suggest that the authors should also report the results obtained from the estimated logistic regression model (and not just refer the reader to the odds ratios shown in Table 3), in order to offer to IJERPH’s readers a comprehensive understanding of the analysed data.

To be clear and concise: I recommend adopting a clear analysis strategy and being rigorous in its application and comprehensible in its description.

We add the B value for the variables and constant and interpretated the results in the discussion section

It is stated in page 6 line 183-188

Present study found that lower educational level, female, no married, and stroke altogether associated with MHD among elderly diabetics with the pseudo-R-squared (Nagelkerke) was 0.790. This finding explained that 79.0% of MHD determinants in this population study of elderly diabetics were influenced by the mentioned factors. The rest 21.0% explained by other factors that did not observe in the study

Reviewer 2 Report

  1. Need more explanations about why you choose binary logistic regression as your analysis method, ex.: a discriminant analysis.
  2. Need more presentation for the data analysis.
  3. How do you explain that diabetics caused MHD or on the contrary?

Author Response

Response to Reviewer 2 Comments

Dear Editors and Reviewers

We thank the editor and reviewers that provide us the importantly substantial review and inputs. We believe these reviews will improve our manuscript; therefore, we are trying the best to revise.

Point 1

Need more explanations about why you choose binary logistic regression as your analysis method, ex.: a discriminant analysis.

Response 1

We add the information in the methods section

Binary logistic regression with the Backward Elimination (Conditional) method was conducted to acquire the regression model since the dependent variable scale was nominal.

(Page 3, line 107-109)

Point 2

Need more presentation for the data analysis.

Response 2

We conducted Backward conditional stepwise Binary logistic regression and present the results in table 3.

Point 3

How do you explain that diabetics caused MHD or on the contrary?

Response 3

We explain in the introduction as well as the addition in the discussion

MHD in diabetics may decrease quality of life[15], poor self-care management[16], increase disability[17], cardiovascular mortality risk[18] and all-cause mortality risk[19] On the other side, diabetes is a risk factor for MHD[20].

In the discssion section: (pag6 line 167-181)

Evidence shows that diabetes mellitus is reciprocally associated with MHD and coincides as comorbidity[38]. Depression is the common MHD that is discussed as a risk factor of DM;[20] however, the underlying mechanism is still unclear. Chronic stress induces immune dysfunction through the hypothalamus-pituitary-adrenal axis and the sympathetic nervous system, caused hypercortisolemia and promotes insulin resistance, visceral obesity, and leads to metabolic syndrome and DM[38]. Furthermore, chronic stress increases the production of inflammatory cytokines. High inflammatory cytokines interact with the function of the pancreatic β-cells, induce insulin resistance, and promote the appearance of type 2 diabetes mellitus[38]. On the other hand, pro-inflammatory cytokines have been reported to influence pathophysiological domains that characterize depression, including neurotransmitter metabolism, neuroendocrine function, synaptic plasticity, and behavior[39]. This association suggested that both stress and inflammation stimulate depression and diabetes mellitus[38]. Chronic stress and inflammation processes, as well as the physical and psychosocial changes that are common in the elderly population, affect both mental health and diabetes in the elderly[30].

Reviewer 3 Report

This article examines the prevalence rates of “mental health disorders” as well as various potential risk factors/comorbidities for a sample of adults aged 60 or over with diabetes in Indonesia.  The study possesses a number of notable strengths, such as access to a large, national database, the recency of data collection, and the inclusion of multiple demographic and health-related variables. It is also a topic of significant public health import. The consideration of the following comments would greatly strengthen the paper:

-The introduction would benefit from clarification of important points. For example:  a)The authors state, “Mental health disorder (MHD) is the most frequent comorbid for DM with a prevalence of 28% globally;” This statement seems to indicate that MHDs have a higher comorbidity with DMs than any other disorder (including hypertension and diabetes). Was that their intent? b) "Previous mental disorder conditions such as…are the underlying comorbid for DM patients with MHD…” Can the authors clarify what is meant here?   

-Greater detail regarding how Mental Health Disorder (MHD) is defined and how MHD was determined would benefit the paper. Which mental health disorders were included in the term?  How did the authors determine that the answers to the questions reached threshold for a “disorder”?  Please also include psychometric support.                         

-Some theoretical rationale/framework for the selection of the “risk factors” would be helpful. 

- The presence of diabetes was determined by the response to a single question related to the receipt of a diagnosis of DM by a doctor. The limitations of this approach could be identified. In addition, it is unclear if other studies utilize the same methodology for determining the presence of DM; it is possible that differential methods for ascertaining DM may affect the interpretation of comparisons with these studies.

Author Response

Response to Reviewer 3 Comments

Dear Editors and Reviewers

We thank the editor and reviewers that provide us the importantly substantial review and inputs. We believe these reviews will improve our manuscript; therefore, we are trying the best to revise.

General comments

This article examines the prevalence rates of “mental health disorders” as well as various potential risk factors/comorbidities for a sample of adults aged 60 or over with diabetes in Indonesia.  The study possesses a number of notable strengths, such as access to a large, national database, the recency of data collection, and the inclusion of multiple demographic and health-related variables. It is also a topic of significant public health import. The consideration of the following comments would greatly strengthen the paper:

Point 1

The introduction would benefit from clarification of important points. For example:  a)The authors state, “Mental health disorder (MHD) is the most frequent comorbid for DM with a prevalence of 28% globally;” This statement seems to indicate that MHDs have a higher comorbidity with DMs than any other disorder (including hypertension and diabetes). Was that their intent? b) "Previous mental disorder conditions such as…are the underlying comorbid for DM patients with MHD…” Can the authors clarify what is meant here?   

Response 1

We changed the statement into Mental health disorder (MHD) is the common comorbid for DM with a prevalence of 28% globally; to clarify this important point.

The second statement changed into: Mental disorders such as generalized anxiety disorder (GAD), major depressive disorder (MDD), bipolar disorder, and eating disorders are common mental disorders in DM patients

Mental health disorder (MHD) is the common comorbid for DM with a prevalence of 28% globally; females tend to be higher than males, i.e., 34% and 23%, respectively[7–10]. Mental disorders such as generalized anxiety disorder (GAD), major depressive disorder (MDD), bipolar disorder, and eating disorders are common mental disorders in DM patients[11–14]. MHD in diabetics may decrease quality of life[15], poor self-care management[16], increase disability[17], cardiovascular mortality risk[18] and all-cause mortality risk[19]. On the other side, diabetes is a risk factor for MHD[20]. (line 40-47)

Point 2

Greater detail regarding how Mental Health Disorder (MHD) is defined and how MHD was determined would benefit the paper. Which mental health disorders were included in the term?  How did the authors determine that the answers to the questions reached threshold for a “disorder”?  Please also include psychometric support.

Response 2

We add the explanation in the methods section. (line 92-102)

. MHD status was determined by the WHO self-reporting questionnaire-20 (SRQ-20),[33–35] as acquired from RISKESDAS 2018 data. SRQ-20 is the tool used to measure common mental disorder symptoms[33–35]. SRQ-20 consists of 20 questions regarding the prevalence of somatic, cognitive, and emotional symptoms over the past 30 days, measure 0=No and 1=Yes[33–35]. RISKESDAS 2018 refers to the previous study that validated SRQ-20 in the Indonesian population[35]. The study determined MHD with the cut-off point ≥6; positive predictive value=70% and negative predictive value=92%[35]. Secondary data were also acquired from RISKESDAS 2018 that involved age, sex, urban-rural residence status, marital status, educational level, employment status, obesity, hypertension, heart disease, stroke, family history of MHD, and duration of DM.

Point 3

Some theoretical rationale/framework for the selection of the “risk factors” would be helpful. 

Response 3 

We add statement to strenghthen the selection of the risk factors.(line 47 – 69)

In general population studies, younger diabetics are more likely to get MHD[8]. Another study reported that elderly diabetics are more likely to get MHD, with the increased risk of other factors [11]. The previous study also concluded that MHD is more likely to occur in females with no formal education, current alcohol abusers, type 1 DM, longer duration of DM, a chronic complication of DM, and other comorbidities among elderly diabetics patients[21]. The other previous studies concern the association of MHD diabetic comorbidity with genetic and family history[22–25] and obesity[26–28].

The frequent coexistence of mental health conditions in elderly diabetics should be of concern[29]. The mechanisms of psychiatric illness involving brain-derived neurotrophic factors, insulin resistance, and inflammatory cytokines could be due to the pathogenesis of DM and several psychiatric illnesses in the elderly[29]. Physical and psychosocial changes affect both mental health and diabetes in the elderly[30]. Diabetic complications such as retinopathy, nephropathy, neuropathy, coronary artery disease, and cerebrovascular disease were also associated with the poor mental health status in elderly diabetics [31]. The other study also concluded that overweight status, poor physical capabilities, low activity level, and diabetic complications were risk factors for depression in elderly diabetic patients[32].

Point 4

The presence of diabetes was determined by the response to a single question related to the receipt of a diagnosis of DM by a doctor. The limitations of this approach could be identified. In addition, it is unclear if other studies utilize the same methodology for determining the presence of DM; it is possible that differential methods for ascertaining DM may affect the interpretation of comparisons with these studies.

Response 4

We revised the Diabetes status as stated in the RISKESDAS 2018 as follows (line 81-84):

Diabetics' status was determined by fasting blood glucose level ≥ 126 mg/dL or 2 hours postprandial and random blood glucose level ≥ 200 mg/dL or previously had been diagnosed by a doctor. Blood glucose levels were measured using Accu-Check Performa (Roche, Swiss).

Reviewer 4 Report

This research focuses on a very interesting study field, the prevalence and risk factors of mental health disorders among diabetic elderly  in one poeple specific (Indonesia). However, the manuscript has several weaknesses: (i) the specific relevance of the study is not included in the introduction, (ii) the diagnosis of diabetes mellitus is not reliable, (iii) it is not specified what type of mental disorders and how they were diagnosed, (iv) the findings are not analyzed considering the reliability of the diagnosis.

Major comments

  1. Introduction.
  • The scientific support for the study on the prevalence and trend of mental disorders in diabetic elderly should be included. In this sense, it is important to highlight that the biological changes of aging linked to diabetes that could increase the risk of mental disorders.
  1. Method:
  • The reliability of the diagnosis of diabetes is unreliable. What is the certainty that people understand the question regarding the diagnosis of diabetes mellitus?
  • Another important limitation is whether diabetes mellitus was controlled and what drugs were they taking.
  • The major limitation of the study is that the authors do not include questions or instruments for the detection of mental disorders.
  1. Discussion
  • The comparison of the findings does not consider the reliability of the diagnosis, since the authors do not include the instrument or diagnostic criteria for mental disorders. In this sense, it is not justified to contrast their findings with other studies, whose dysnostic was perhaps carried out with other criteria or instruments.
  • Some included analyzes are speculations, as the authors did not measure biological markers of inflammation.
  • It is necessary for the analysis of results to have a gerontological approach.

In overal, I consider that the study does not have sufficient scientific support to justify it, especially since the criteria, questions or instruments for the diagnosis of -Mental Health Disorders- are not included.

Author Response

Response to Reviewer 4 Comments

Dear Editors and Reviewers

We thank the editor and reviewers that provide us the importantly substantial review and inputs. We believe these reviews will improve our manuscript; therefore, we are trying the best to revise.

General comment

This research focuses on a very interesting study field, the prevalence and risk factors of mental health disorders among diabetic elderly in one poeple specific (Indonesia). However, the manuscript has several weaknesses: (i) the specific relevance of the study is not included in the introduction, (ii) the diagnosis of diabetes mellitus is not reliable, (iii) it is not specified what type of mental disorders and how they were diagnosed, (iv) the findings are not analyzed considering the reliability of the diagnosis.

Major comments

Point 1

  1. Introduction.
  • The scientific support for the study on the prevalence and trend of mental disorders in diabetic elderly should be included. In this sense, it is important to highlight that the biological changes of aging linked to diabetes that could increase the risk of mental disorders.

Response 1

We revised the introduction and add some information regarding stress and inflammation processes, stated as follows (line 55-64)

The frequent coexistence of mental health conditions in elderly diabetics should be of concern[29]. The mechanisms of psychiatric illness involving brain-derived neurotrophic factors, insulin resistance, and inflammatory cytokines could be due to the pathogenesis of DM and several psychiatric illnesses in the elderly[29]. Physical and psychosocial changes affect both mental health and diabetes in the elderly[30]. Diabetic complications such as retinopathy, nephropathy, neuropathy, coronary artery disease, and cerebrovascular disease were also associated with the poor mental health status in elderly diabetics [31]. The other study also concluded that overweight status, poor physical capabilities, low activity level, and diabetic complications were risk factors for depression in elderly diabetic patients

Point 2

  1. Method:
  • The reliability of the diagnosis of diabetes is unreliable. What is the certainty that people understand the question regarding the diagnosis of diabetes mellitus?

Response 2 a

We revised the statements as stated in the Riskesdas 2018. (line 81-83)

Diabetics’ status was determined by fasting blood glucose level ≥ 126 mg/dL or 2 hours postprandial and random blood glucose level ≥ 200 mg/dL or previously had been diagnosed by a doctor. Blood glucose levels were measured using Accu-Check Performa (Roche, Swiss)

  • Another important limitation is whether diabetes mellitus was controlled and what drugs were they taking.

Response 2 b

We add this explanation as our study limitation (line 245-248)

Our study limitation is the absence observation of diabetic medication status and glycemic control[29, 32]. Glycemic control and use of certain oral medication is concluded that related to the mental health condition in elderly diabetics[29, 32].

  • The major limitation of the study is that the authors do not include questions or instruments for the detection of mental disorders.

Response 2 c

We revised and add the explanation in the methods (line 92-99). We also attched the questionnaire in the supplementation file

MHD status was determined by the WHO self-reporting questionnaire-20 (SRQ-20),[33–35] as acquired from RISKESDAS 2018 data. SRQ-20 is the tool that was used to measure symptoms of common mental disorder[33–35]. SRQ-20 consists of 20 questions regarding the prevalence of somatic, cognitive, and emotional symptoms over the past 30 days, measure 0=No and 1=Yes[33–35]. RISKESDAS 2018 refers to the previous study that validated SRQ-20 in the Indonesian population[35]. This study determined MHD with the cut-off point ≥6; positive predictive value=70% and negative predictive value=92%[35]

Point 3

  1. Discussion
  • The comparison of the findings does not consider the reliability of the diagnosis, since the authors do not include the instrument or diagnostic criteria for mental disorders. In this sense, it is not justified to contrast their findings with other studies, whose dysnostic was perhaps carried out with other criteria or instruments.

Response 3 a

We revised the reliability of the diagnosis and explain in the discussion section (159-162)

Depression determination methods also influence the prevalence ratio; self-reported methods tend to had a higher prevalence (30%) than clinical diagnosis assessment (22%)[8]. The current study utilized self-reported methods using WHO-SRQ-20[33–35].; however, it found a lower number of prevalent compared to the previous review[8]

  • Some included analyzes are speculations, as the authors did not measure biological markers of inflammation.

Response 3 b

We add this in the limitation of the study (line 247-255)

Previous studies involved stress and epigenetics as a predictor of MHD in the general population[23]. Candidate genes were also studied and revealed that APOE, BDNF, SLC6A4 polymorphisms related to MHD in the general population[24]. Other studies revealed the contribution of inflammatory markers to mental health. Interleukin (IL)-1β, IL-6, IL-10, monocyte chemoattractant protein-1, tumor necrosis factor-alpha, C-reactive protein, and phospholipase A2 contributed to the depression[25], which is also associated with type 2 diabetes[49]. However, our study did not involve biological and genetic markers to elucidate the understanding of these mechanisms.

  • It is necessary for the analysis of results to have a gerontological approach.

Response 3 c

We add this approach to explain in the discussion section (line 179-181)

Chronic stress and inflammation proccesss as well as the physical and psychosocial changes that common in elderly population, affect both mental health and diabetes in elderly[30].

Point 4

In overal, I consider that the study does not have sufficient scientific support to justify it, especially since the criteria, questions or instruments for the diagnosis of -Mental Health Disorders- are not included

Response 4

We add and included the the criteria, questions or instruments for the diagnosis of Mental Health Disorders as stated in Response 2

Reviewer 5 Report

The authors conducted a study to evaluate the prevalence of mental disorders among elderly diabetics in Indonesia, as well as risk factors, using data from a national health survey. Strenghts of this study are the fact that the sample is not a convenience sample but it is representative of the population. A limitation is that mental disorders were self reported. 

  1. The manuscript in its current form is a bit difficult to read as it contains several errors (for instance missing verbs, subjects, other typos or unclear phrasing). See for instance at page 2, lines 40-42: "Previous mental disorder conditions such as generalized anxiety disorder (GAD), major depressive disorder (MDD), bipolar disorder, and eating disorders are the underlying comorbid for DM patients with MHD". 
  2. The introduction should provide additional information on previous literature on this topic (for instance at page 2, lines 46-48, there is no information on the type of participants/sample included by previous studies, as well as their limitations).
  3. Additional explanation on the questionnaire used to defined MHD status would be useful as at present it is not clear which disorders (although self-reported) might be included in this definition.
  4. In the Statistical Analysis paragraph, the authors state: "Parameters that had p-value <0.25 then involved in the multivariate analysis using binary logistic regression". The authors should explain the rationale underlying the definition of this threshold. Also, the word "then" is unclear (see comment 1). The logistic regression model should be better described.
  5. Please add a sentence to describe how prevalence odds ratios were calculated.
  6. In Table 3, hypertension is mispelled.

Author Response

Response to Reviewer 5 Comments

Dear Editors and Reviewers

We thank the editor and reviewers that provide us the importantly substantial review and inputs. We believe these reviews will improve our manuscript; therefore, we are trying the best to revise.

General comments

The authors conducted a study to evaluate the prevalence of mental disorders among elderly diabetics in Indonesia, as well as risk factors, using data from a national health survey. Strenghts of this study are the fact that the sample is not a convenience sample but it is representative of the population. A limitation is that mental disorders were self reported. 

Point 1

The manuscript in its current form is a bit difficult to read as it contains several errors (for instance missing verbs, subjects, other typos or unclear phrasing). See for instance at page 2, lines 40-42: "Previous mental disorder conditions such as generalized anxiety disorder (GAD), major depressive disorder (MDD), bipolar disorder, and eating disorders are the underlying comorbid for DM patients with MHD". 

Response 1

We make changes in difficult stentences regarding some errors

Mental disorders such as generalized anxiety disorder (GAD), major depressive disorder (MDD), bipolar disorder, and eating disorders are common mental disorders in DM patients[11–14]

Point 2

The introduction should provide additional information on previous literature on this topic (for instance at page 2, lines 46-48, there is no information on the type of participants/sample included by previous studies, as well as their limitations).

Response 2

We revised and add study population

In general population studies, younger diabetics are more likely to get MHD[8] while another study reported that elderly diabetics more likely to get MHD, with the increased risk of other factors present[11]. The previous study also concluded that MHD is more likely to occur in females, no formal education, current alcohol abusers, type 1 DM, longer duration of DM, chronic complication of DM, and other comorbidities among elderly diabetics patients[21]. The other previous studies concern the association of MHD diabetic comorbidity with genetic and family history[22–25] and obesity[26–28].

Point 3

Additional explanation on the questionnaire used to defined MHD status would be useful as at present it is not clear which disorders (although self-reported) might be included in this definition.

Response 3

We revised and add more explanation about the term of MHD

MHD status was determined by the WHO self-reporting questionnaire-20 (SRQ-20),[33–35] as acquired from RISKESDAS 2018 data. SRQ-20 is the tool that was used to measure symptoms of common mental disorder[33–35]. SRQ-20 consists of 20 questions regarding the prevalence of somatic, cognitive, and emotional symptoms over the past 30 days, measure 0=No and 1=Yes[33–35]. RISKESDAS 2018 refers to the previous study that validated SRQ-20 in the Indonesian population[35]. This study determined MHD with the cut-off point ≥6; positive predictive value=70% and negative predictive value=92%[35]

Point 4

In the Statistical Analysis paragraph, the authors state: "Parameters that had p-value <0.25 then involved in the multivariate analysis using binary logistic regression". The authors should explain the rationale underlying the definition of this threshold. Also, the word "then" is unclear (see comment 1). The logistic regression model should be better described.

Response 4

We revised and perform re-analysis using Bacward conditional step wise, Binary logistic regression.

Binary logistic regression with the Backward Elimination (Conditional) method was conducted to acquire the regression model since the dependent variable scale was nominal.

Point 5

Please add a sentence to describe how prevalence odds ratios were calculated.

Response 5

We add information regarding POR

We presented prevalence odds ratio (POR) for cross-sectional study as formulated in the previous study[36]

Point 6

In Table 3, hypertension is mispelled.

Response 6

We removed hypertension since it did not appear in the model (after perform re-analysis).

Round 2

Reviewer 1 Report

Some Remarks 

On page 3 (line 108) authors state that “... the dependent variable scale was nominal." Logistic regression is the most important model for a binary response variable.

Authors must specify which criterion was used to perform stepwise model selection and what value it assumed.

Author Response

Response to Reviewer 1 Comments

Thank you very much for continuing to give us constructive inputs to improve our manuscript. We try to address the comment to refine our manuscript.

Comment

Some Remarks 

On page 3 (line 108) authors state that “... the dependent variable scale was nominal." Logistic regression is the most important model for a binary response variable.

Authors must specify which criterion was used to perform stepwise model selection and what value it assumed.

Response:

We revised on page 3 line 112-114

The dependent variable was MHD status that categorized as “Yes” if meet the criterion and “No” if not.

Reviewer 4 Report

The authors have corrected the manuscript considering all the comments.

Author Response

Response to Reviewer 1 Comments

Comment

The authors have corrected the manuscript considering all the comments.

Response:

Thank you very much for constructive inputs to improve our manuscript

Reviewer 5 Report

The authors addressed most of the raised points (although they replaced an arbitrary p-value threshold for which I had asked to support the choice, with a method which is considered to be highly problematic, see for instance: https://journalofbigdata.springeropen.com/articles/10.1186/s40537-018-0143-6)

In my opinion, a careful revision of English language is needed before publication as the meaning of several sentences is unclear. 

Author Response

Response to Reviewer 1 Comments

Thank you very much for continuing to give us constructive inputs to improve our manuscript. We try to address the comment to refine our manuscript.

Comments

Poin 1

The authors addressed most of the raised points (although they replaced an arbitrary p-value threshold for which I had asked to support the choice, with a method which is considered to be highly problematic, see for instance: https://journalofbigdata.springeropen.com/articles/10.1186/s40537-018-0143-6)

Response 1

We add the explanation in the limitation regarding the analysis method

Page 8 line 268-270

Furthermore, regarding the statistical analysis, a backward stepwise binary logistic regression was chosen as an efficient method for the extensive data; however, there are some restrictions regarding this method [51].

Poin 2

In my opinion, a careful revision of English language is needed before publication as the meaning of several sentences is unclear. 

Response 2

We request the professional english editing to fix grammatical errors and unclear sentences.
